# Recent Insights on the Role of Nuclear Receptors in Alzheimer’s Disease: Mechanisms and Therapeutic Application

**DOI:** 10.3390/ijms26031207

**Published:** 2025-01-30

**Authors:** Xiaoxiao Shan, Dawei Li, Huihui Yin, Wenwen Tao, Lele Zhou, Yu Gao, Chengjie Xing, Caiyun Zhang

**Affiliations:** 1Anhui Academy of Chinese Medicine, Anhui University of Chinese Medicine, Hefei 230012, China; shanxiaoxiao2022@126.com (X.S.); 15240181837@163.com (D.L.); yhh@stu.cpu.edu.cn (H.Y.); taoww0506@163.com (W.T.); zhoule6112@163.com (L.Z.); gaoyu0210@163.com (Y.G.); xingxinyu304@163.com (C.X.); 2Center for Xin’an Medicine and Modernization of Traditional Chinese Medicine of IHM, Grand Health Research Institute of Hefei Comprehensive National Science Center, Anhui University of Chinese Medicine, Hefei 230012, China; 3Engineering Technology Research Center of Modernized Pharmaceutics, Anhui Education Department (AUCM), Hefei 230012, China; 4School of Pharmacy, Institute of Pharmacokinetics, Anhui University of Chinese Medicine, Hefei 230012, China; 5Anhui Genuine Chinese Medicinal Materials Quality Improvement Collaborative Innovation Center, Hefei 230012, China; 6Anhui Province Key Laboratory of Pharmaceutical Preparation Technology and Application, Anhui University of Chinese Medicine, Hefei 230012, China

**Keywords:** Alzheimer’s disease, nuclear receptors, β-amyloid, mechanisms, therapeutic target

## Abstract

Nuclear receptors (NRs) are ligand-activated transcription factors that regulate a broad array of biological processes, including inflammation, lipid metabolism, cell proliferation, and apoptosis. Among the diverse family of NRs, peroxisome proliferator-activated receptors (PPARs), estrogen receptor (ER), liver X receptor (LXR), farnesoid X receptor (FXR), retinoid X receptor (RXR), and aryl hydrocarbon receptor (AhR) have garnered significant attention for their roles in neurodegenerative diseases, particularly Alzheimer’s disease (AD). NRs influence the pathophysiology of AD through mechanisms such as modulation of amyloid-beta (Aβ) deposition, regulation of inflammatory pathways, and improvement of neuronal function. However, the dual role of NRs in AD progression, where some receptors may exacerbate the disease while others offer therapeutic potential, presents a critical challenge for their application in AD treatment. This review explores the functional diversity of NRs, highlighting their involvement in AD-related processes and discussing the therapeutic prospects of NR-targeting strategies. Furthermore, the key challenges, including the necessity for the precise identification of beneficial NRs, detailed structural analysis through molecular dynamics simulations, and further investigation of NR mechanisms in AD, such as tau pathology and autophagy, are also discussed. Collectively, continued research is essential to clarify the role of NRs in AD, ultimately facilitating their potential use in the diagnosis, prevention, and treatment of AD.

## 1. Introduction

Alzheimer’s disease (AD) is a major neurodegenerative disorder that primarily affects the elderly, leading to severe cognitive decline and functional impairment [1]. Pathologically, AD is characterized by a progressive accumulation of β-amyloid (Aβ) plaques, tau protein hyperphosphorylation, neuroinflammation, and neuronal loss [2,3]. The disease manifests in both sporadic and familial forms, each associated with distinct molecular mechanisms and genetic mutations [4]. In sporadic AD, the most common form, the exact causes remain elusive, but several risk factors have been identified, with the apolipoprotein E (*APOE*) ε4 allele being the most significant genetic risk factor. Individuals carrying the APOE ε4 allele exhibit a significantly higher risk of developing AD, with a 4–15-fold increased incidence and an earlier onset compared to those without the allele [5]. *APOE ε4* has been shown to facilitate the accumulation of Aβ plaques and neurofibrillary tangles, hallmarks of AD pathology, by altering Aβ metabolism and impeding its clearance from the brain [6]. Interestingly, homozygous carriers of *APOE ε4* show a greater risk and earlier onset of AD compared to heterozygous carriers, underscoring the role of genetic predisposition in disease development [7]. At the molecular level, one of the primary pathological features of AD is the abnormal accumulation of β-amyloid peptides, particularly Aβ_1–42_. This peptide is derived from the cleavage of amyloid precursor protein (APP) by β-secretase and γ-secretase [8]. Under normal conditions, Aβ is produced in a regulated manner and cleared efficiently. However, in AD, an imbalance between Aβ production and clearance leads to the formation of insoluble Aβ plaques, which aggregate between neurons and disrupt their function [9]. These plaques trigger a cascade of neurotoxic events, including oxidative stress, inflammation, and impaired synaptic function, contributing to the progressive degeneration of neurons and cognitive decline [10]. Tau protein, another critical player in AD pathogenesis, undergoes hyperphosphorylation in AD, leading to the formation of neurofibrillary tangles [11]. These tangles disrupt intracellular transport, contribute to neuronal dysfunction, and are strongly correlated with cognitive deficits in AD patients [12]. The interplay between Aβ deposition and tau hyperphosphorylation accelerates neuronal damage, leading to the characteristic neurodegenerative processes seen in AD [13]. In addition to these molecular hallmarks, other factors, such as mitochondrial dysfunction, impaired autophagy, and neuroinflammation, play a significant role in the pathogenesis of AD [14,15].

Familial forms of AD are less common but are caused by mutations in specific genes, such as *APP*, presenilin 1 (*PSEN1*), and presenilin 2 (*PSEN2*), which are involved in the processing of APP and Aβ [16]. These mutations lead to the overproduction or altered cleavage of Aβ, accelerating plaque formation and the onset of neurodegeneration. However, familial AD accounts for a small fraction of total cases, and the majority of AD cases are sporadic, with multifactorial influences involving genetic, environmental, and lifestyle factors [17]. Despite extensive research into AD’s pathogenesis, therapeutic approaches remain limited [18]. Current treatments primarily aim to alleviate symptoms or slow disease progression but do not address the underlying molecular mechanisms. Therefore, there is an urgent need for novel therapeutic strategies targeting key pathways in AD pathogenesis.

In recent years, nuclear receptors (NRs) have garnered significant attention in the study of AD pathogenesis due to their crucial role in regulating gene expression and the cellular processes that influence neuronal health [19]. By modulating gene expression, metabolism, inflammation, and protein aggregation, NRs offer a promising therapeutic strategy for treating AD. NRs are a family of ligand-activated transcription factors that mediate cellular responses by binding to specific ligands, such as hormones, metabolites, or lipids, and then modulating gene expression. The human genome contains 48 known NRs; these NRs are highly conserved in both function and sequence across species, highlighting their fundamental biological importance [20]. As shown in Figure 1, NRs typically function as dimers, binding to DNA response elements, where they can recruit co-activators or co-repressors to regulate the expression of target genes [21]. This dynamic regulation of gene expression is essential for maintaining cellular homeostasis and responding to environmental signals [22]. In the context of AD, NRs are involved in various critical brain functions, including neurogenesis, synaptic plasticity, neuronal differentiation, and metabolism. They are also key regulators of glycolipid metabolism, which is crucial for brain energy homeostasis [23]. Moreover, NRs have been implicated in modulating processes like inflammation and oxidative stress, both of which contribute to neurodegeneration in AD [24]. The interest in NRs as potential therapeutic targets for AD is based on their capacity to influence these processes at a molecular level. Specific NRs, such as the peroxisome proliferator-activated receptors (PPARs), retinoid X receptor (RXR), and estrogen receptors (ERs), have been shown to affect Aβ metabolism, tau phosphorylation, and neuroinflammatory pathways, all of which are central to AD pathology [25,26]. Additionally, NRs regulate mitochondrial function, which is compromised in AD, and may offer a pathway for protecting neuronal health and preventing the progression of neurodegenerative damage [27]. Thus, the study of NRs in AD offers a promising avenue for uncovering new therapeutic strategies aimed at slowing or reversing the molecular hallmarks of the disease.

Numerous studies on the structures of the DNA-binding domain (DBD) and ligand-binding domain (LBD) of NRs have shown that these receptors are promising targets for treating chronic neurodegenerative diseases [28,29,30]. Research on NR agonists is already underway as a potential treatment for AD. This review explores the functional diversity of NRs and their involvement in AD-related processes, while also discussing the therapeutic potential of strategies that target NRs. In particular, the roles of various NRs, including peroxisome proliferator-activated receptors (PPARs), estrogen receptor (ER), liver X receptor (LXR), farnesoid X receptor (FXR), retinoid X receptor (RXR), and aryl hydrocarbon receptor (AhR), in AD are highlighted. A deeper understanding of the mechanisms by which NRs influence AD could offer new therapeutic targets for the disease.

## 2. Structure and Function of NRs and Their Role in AD Pathogenesis

NRs are a class of ligand-dependent transcription factors that are widely distributed in organisms [31]. Unlike other signaling processes, intracellular NRs can be activated by binding directly to ligands [26]. Since some NRs have ligands that are not yet known or may not exist, they are called ‘orphan NRs’ [32]. Depending on the presence or absence of a ligand, NRs are categorized as classical NRs, which include steroid hormone receptors and non-steroidal hormone receptors [33]. With further study of the physiology and biochemistry of NRs, some ligands for orphan NRs have been discovered. For example, the pregnane X receptor (PXR) was initially considered an orphan NR, but related ligands have since been discovered, such as the RXR, vitamin D receptors (VDR), PPARs, and liver X receptor (LXR). Another category is orphan NRs [34]. In addition, according to ligand DNA-binding properties and dimerization, NRs can be divided into type I NRs, also known as steroid hormone receptors, such as ER, androgen receptor (AR), and glucocorticoid receptor (GR) [35]. Type II NRs, also known as non-steroidal hormone receptors, are heterodimerized with RXR and are characteristically bound to repetitive sequences, including PPAR, LXR, and VDR [36]. Type III NRs homodimerize directly with repetitive sequences, and type IV NRs are mostly in the form of a monomer with the binding site. Most orphan receptors belong to types III and IV [37].

In the pathogenesis of AD, the NRs act as transcription factors that bind directly to ligands and conduct chemical signals from the ligand to the target gene, resulting in changes in gene expression, a process that has been emphasized by structural biologists [25]. A comparison of the cDNA sequences of NRs and their encoded amino acid sequences reveals that NRs belong to a superfamily of transcription factors that are evolutionarily conserved. Typical NRs generally include five functional regulatory regions: A, B, C, D, and E. The A/B region contains at least one ligand-independent transcriptional activation domain (AF-1), which is highly variable in length, ranging from 50 to 500 amino acids [38]. The C region corresponds to the highly conserved DNA-binding domain (DBD), which contains two C4-zinc-like fingers that connect the promoter region near the receptor to the target gene. This region comprises hormone response elements (HREs) or xenobiotics response elements (XREs) [39]. The primary function of the hormone response element (HRE) is to recognize specific DNA sequence motifs, known as hormone response elements, which have enhancer properties on target genes. It binds to these elements to regulate the transcription of effector genes [40]. The largest domain in the nuclear receptors (NRs) is the E domain, also known as the ligand-binding domain (LBD), which is primarily responsible for ligand binding. This domain is highly conserved, ensuring the precise recognition of the selected ligand [40]. LBD also includes a ligand-dependent transcriptional activation domain (AF-2), which is relatively large and complex in function and is generally believed to mediate nuclear receptor dimerization, nuclear localization, transcriptional activation, intramolecular silence, and intermolecular repression [41]. In the DNA-binding region (C region) and ligand-binding region (E region), there are short and unconservative structures called hinge regions, which contain nuclear localization signaling peptides (NLS). Furthermore, the F region is located at the carboxyl terminal, and the sequence of the F region varies greatly among different NRs (Figure 1). Recent studies have found that the occurrence of tumors is related to mutations in the hinge region [42].

As described by Almeida et al., NRs contain a central region DBD with a hormone response element consisting of two zinc ions that maintain the folding of the core DBD; it produces a response by recognizing the correct DNA half-site and inserting it into the groove of the conserved hexamer of the DNA response element [43]. The LBD is a hydrophobic cavity with varying sizes and structures, composed of 12 or 13 α-helices in an antiparallel arrangement. This structure enables it to bind different lipophilic ligands, recognize response elements, and ensure the specificity and selectivity of the interaction [44]. After binding to ligands, NRs are induced and activated to expose different domains and produce a series of conformational changes. Furthermore, NRs specifically bind to the target genes and exert effects [45]. It has been reported that the recombinant nuclear receptor related protein 1 (Nurr1) monomer was directly interacted with p65 on the response elements of nuclear factor kappa-B (NF-κB) during the polymerization process to recruit the CoREST co-suppressor complex, thereby inhibiting the expression of pro-inflammatory genes regulated by NF-κB and reducing neuroinflammation in AD [46]. Additionally, considerable evidence suggests that NRs can directly or indirectly regulate the expression of *APOE*, *Aβ*, *NF-κB*, *ABCA1,* and other genes through genomic effects [47]. Therefore, NRs represent promising therapeutic targets for AD, playing a key role in its onset and progression through various mechanisms, including modulation of gene expression, regulation of neuroinflammation, lipid metabolism, and Aβ deposition.

## 3. The Pathogenesis of AD and the Mechanistic Involvement of NRs

### 3.1. The Involvement of NRs in Aβ Regulation

The description of the clinical symptoms of AD was first defined in the early 1900s [48]. However, it is only in the last two decades that researchers have opened the era of identifying biomarkers of AD, proposing that abnormal processing of Aβ drives abnormal aggregation of tau proteins [49]. Removal of Aβ deposition in the brain is one of the most popular strategies to ameliorate the pathological process of AD. Aβ peptides are derived from *APP*, which undergoes cleavage by β-secretase (*BACE1*) and γ-secretase to release Aβ fragments, primarily Aβ_1–40_ and Aβ_1–42_. The longer Aβ_1–42_ form is particularly prone to aggregation and is considered more toxic to neurons [50]. In a healthy brain, Aβ is produced at low levels and cleared efficiently via mechanisms involving enzymes such as neprilysin and insulin-degrading enzyme, as well as through the glymphatic system. In AD, there is an imbalance between the overproduction of Aβ and its impaired clearance, leading to its accumulation [50]. The accumulated Aβ forms senile plaques, which are extracellular deposits that disrupt synaptic function, cause oxidative stress, and induce inflammation, all contributing to neuronal damage. Soluble oligomers of Aβ (small aggregates of Aβ) are believed to be the most neurotoxic form, disrupting synaptic plasticity, impairing neuronal signaling, and contributing to neurodegeneration [51]. These oligomers can also activate microglia and astrocytes, exacerbating neuroinflammation.

Studies have reported that NRs can induce Aβ removal or reduce Aβ production by stimulating the expression of the relevant DNA [52,53]. Jiang et al. found that GW2576, an LXR agonist, could stimulate the degradation of Aβ in the brain of aged Tg3965 mice, with a significant reduction in Aβ levels [53]. Furthermore, this action was dependent on the LXR target genes *APOE* and *ABCA1*, suggesting that LXR agonists may be a novel approach for the treatment of AD. Additionally, NRs can inhibit the production of Aβ by regulating APP. Blondrath et al. demonstrated that receptor interacting protein 140 (RIP140) is significantly enriched in the cerebral cortex and hippocampus of AD rats compared to healthy controls and plays a key role in regulating the transcription of genes implicated in AD [54]. RIP140 acts as a critical cofactor for PPARγ, and activation of PPARγ has been shown to reduce Aβ levels both in vitro and in animal models of AD by downregulating the transcription of *BACE1*, a key enzyme involved in Aβ production. The findings further suggest that RIP140 reduces Aβ production in neuroblastoma cells through PPARγ-dependent mechanisms, leading to decreased expression of *APP* cleavage enzymes. Additionally, RIP140 has been identified as a coactivator of NF-κB in macrophages [55], highlighting its multifaceted role in regulating Aβ production and offering potential therapeutic pathways to mitigate the progression of AD [56].

### 3.2. The Involvement of NRs in Regulating Inflammation

Neuroinflammation is an indispensable key link in the upstream of AD pathogenesis [57]. Although neuroinflammation is a natural protective response to injury or disease, in the context of AD it becomes chronic, sustained, and maladaptive, contributing to neuronal damage, synaptic dysfunction, and disease progression [58]. The neuroinflammation in AD mainly has the following aspects.

#### 3.2.1. Activation of Microglia

Microglia, the resident immune cells of the brain, play a pivotal role in the neuroinflammatory response in AD [59]. In healthy conditions, microglia are involved in maintaining homeostasis, clearing cellular debris, and protecting neurons from pathogens. However, in AD, the accumulation of toxic substances, such as Aβ plaques and hyperphosphorylated tau, triggers the activation of microglia [60]. Activated microglia undergo a transformation from a resting, surveillance state to a highly reactive state, in which they release a wide range of pro-inflammatory cytokines, chemokines, and reactive oxygen species (ROS). This chronic activation of microglia has several detrimental consequences [61]. Notably, the production of pro-inflammatory cytokines can exacerbate neuronal injury, impair Aβ clearance, and enhance tau phosphorylation [62].

#### 3.2.2. Astrocyte Dysfunction and Reactive Gliosis

Astrocytes, another major cell type involved in the brain’s immune response, also contribute to neuroinflammation in AD [63]. Under normal circumstances, astrocytes maintain the integrity of the blood–brain barrier, regulate neurotransmitter levels, and support neuronal function [64]. In AD, however, astrocytes become reactive, undergoing a process called reactive gliosis, in which they increase in size, alter their morphology, and release pro-inflammatory factors. Reactive astrocytes release cytokines, such as interleukin-6 (IL-6) and transforming growth factor-β (TGF-β), and upregulate signaling pathways that contribute to neuroinflammation [65]. These activated astrocytes can impair neuronal function, disrupt synaptic signaling, and even exacerbate microglial activation. Additionally, they may contribute to the formation of gliosis-associated scars around Aβ plaques and tau tangles, further limiting neuronal regeneration and repair in affected brain regions [66].

#### 3.2.3. Inflammatory Mediators and Their Impact on Neurons

Chronic neuroinflammation has a direct toxic effect on neurons by disrupting synaptic function, promoting oxidative damage, and impairing cellular homeostasis [67]. For instance, pro-inflammatory cytokines can impair the integrity of the blood–brain barrier, allowing the infiltration of peripheral immune cells into the brain and contributing to further inflammation [68]. ROS and nitric oxide (NO), which are produced during the activation of microglia and astrocytes, can induce oxidative stress, leading to the peroxidation of lipids, protein damage, and mitochondrial dysfunction [69]. This oxidative damage accelerates neuronal death and contributes to the cognitive decline characteristic of AD.

#### 3.2.4. The Vicious Cycle of Neuroinflammation

Neuroinflammation in AD is not a static process but rather a vicious cycle in which inflammatory responses propagate and worsen disease pathology [70,71]. The deposition of Aβ in the brain not only triggers the activation of microglia and astrocytes but also creates a chronic inflammatory environment that further promotes Aβ aggregation. As Aβ accumulates, it can bind to microglial receptors such as TREM2 and CD36, leading to the activation of signaling pathways that enhance inflammation and Aβ deposition [72]. Similarly, hyperphosphorylated tau has been shown to have pro-inflammatory effects, potentially inducing the activation of microglia and astrocytes. Tau aggregates may even spread in a prion-like fashion from one neuron to another, further propagating neuroinflammation across brain regions [73]. This inflammatory cycle accelerates synaptic loss, neuronal degeneration, and cognitive dysfunction, amplifying the disease process [74]. Therefore, targeting neuroinflammation represents a promising therapeutic approach in the fight against AD, with the goal of breaking the inflammatory cycle and preventing the progression of disease.

#### 3.2.5. NRs Involved in the Neuroinflammation in AD

Given the central role of neuroinflammation in AD, targeting the inflammatory pathways associated with microglia and astrocyte activation has become an area of active research. PPARs and LXRs are NRs involved in regulating inflammation and lipid metabolism. Activation of these receptors has been shown to reduce neuroinflammation, improve Aβ clearance, and potentially slow the progression of AD [75,76]. Qu et al. found that the elevated expression levels of orphan NR tailless-like protein (TLX) were accompanied by the upregulation of Sirtuin 1 (SIRT1) and downregulation of inflammatory cytokines, suggesting that TLX can block the development of AD by regulating the SIRT1/NF-κB pathway, promoting hippocampal neurogenesis, and inhibiting neuroinflammatory responses [77]. It has also been demonstrated that PPARs can bind to the p65 subunit of NF-κB, thereby exerting anti-inflammatory effects [78]. Similarly, both estrogen receptors (ERs) and aryl hydrocarbon receptors (AhRs) can inhibit the activation of NF-κB, thereby reducing inflammatory responses [79].

### 3.3. The Involvement of NRs in Regulating Metabolism

Metabolic dysregulation is increasingly recognized as a central feature in AD pathogenesis, contributing to the progressive neurodegeneration that underpins the disease [80]. The brain is highly energy-demanding, and its proper function relies on efficient metabolism to support neuronal activity, synaptic plasticity, and cellular maintenance [81]. In AD, however, several metabolic pathways are disrupted, leading to neuronal dysfunction, oxidative stress, and ultimately cognitive decline [82]. Understanding and regulating metabolism in the context of AD is critical for identifying potential therapeutic strategies to alleviate or slow disease progression.

#### 3.3.1. Regulating the Gut–Brain Axis

The interaction between the gut and the brain, often referred to as the gut–brain axis, has become a critical focus in understanding the pathophysiology of various neurodegenerative diseases, including AD [83]. Research has increasingly pointed to the significant role of the gut microbiota in influencing brain function, offering insights into how the development of AD may be not only a result of direct neurodegenerative processes but also an outcome of disturbances in gut health [84,85]. The gut–brain axis is a complex communication network that involves multiple pathways, including the vagus nerve and the hypothalamic–pituitary–adrenocortical (HPA) axis [86]. The vagus nerve, which transmits signals from the gut to the brain, is one of the primary mediators of this relationship. Research has demonstrated that disturbances in the gut microbiome can influence neural activity through this pathway, potentially triggering or exacerbating neuroinflammation and promoting the aggregation of toxic proteins such as Aβ and tau in the brain [87]. Tryptophan, an essential amino acid derived from the diet, is metabolized in the gut by the microbiota into several bioactive metabolites, including kynurenine and indoles. These metabolites act on AhR, a class of nuclear receptors involved in regulating immune responses and maintaining gut homeostasis [88]. Recent studies have demonstrated that the metabolites generated by the gut microbiota, either through dietary components or microbial activity, can ameliorate the progression of neurodegenerative diseases such as AD by promoting the activation of AhR [89,90]. For example, metabolites such as indole-3-propionic acid (IPA) have been shown to exert neuroprotective effects by inducing SIRT1 activity, leading to a reduction in neuroinflammation and enhancing synaptic function [91]. This interaction between the gut microbiota and AhR represents a promising therapeutic avenue for the treatment or prevention of AD, suggesting that modulation of the microbiota through dietary or pharmacological interventions could potentially mitigate the neuroinflammatory processes that contribute to the disease.

#### 3.3.2. Regulating Bile Acid Metabolism

Bile acids are synthesized in hepatocytes and are the main pathway for cholesterol metabolism in the body [92]. It has been shown that bile acids play an important role in lipid digestion and absorption in vivo through activating receptors such as FXR [93]. Approximately 25% of human cholesterol is present in the brain and is an important component of neural development and neuronal composition [94]. It has been reported that FXR receptors are highly expressed in response to elevated bile acid concentrations [95]. Additionally, a comparative clinical study of healthy populations and patients with AD has found that the lithocholic acid levels in plasma were significantly higher in patients with AD than in healthy populations. This implies that bile acids could be used as one of the diagnostic criteria for AD [96]. In order to investigate the mechanism of action of bile acids in AD, Wu et al. applied Aβ_1–42_ to establish an AD model, and gave the G protein-coupled bile acid receptor 1 (GPBAR1) agonist INT-777 as a treatment [95]. The results suggested that INT-777 significantly improved the cognitive deficits, inhibited the upregulation of NF-κB p65, and attenuated neuroinflammatory responses in AD mice, and the results indicated that the GPBAR1 agonist may be a potential target for the treatment of AD [95].

### 3.4. The Involvement of NRs in Improving Neuronal Dysfunction

#### 3.4.1. The Neuronal Dysfunction in AD

Neuronal dysfunction is a hallmark feature of AD; it manifests as a progressive decline in cognitive abilities, including memory, learning, and problem solving [97]. The dysfunction of neurons in AD is multifactorial, involving a complex interplay of genetic, environmental, and metabolic factors that lead to both structural and functional impairments in the brain [98]. Over time, these dysfunctions contribute to the clinical symptoms of dementia, which include memory loss, language difficulties, and impaired executive function. In short, neuronal dysfunction in AD is a multifaceted process, involving the accumulation of Aβ plaques and tau tangles, mitochondrial dysfunction, neuroinflammation, calcium dysregulation, and loss of synaptic integrity [99]. These events not only disrupt the normal functioning of neurons but also contribute to the progressive nature of the disease. Understanding the mechanisms of neuronal dysfunction in AD is crucial for developing therapeutic strategies aimed at slowing or halting the progression of the disease and improving the quality of life for affected individuals.

#### 3.4.2. The Role of NRs in the Improvement of Neuronal Function in AD

The activation of NRs facilitates neuronal function and development, which could help us to identify new therapeutic targets. For instance, LXR agonists can modify DNA methylation patterns and lower the methylation levels of genes involved in synaptic function and neurogenesis. This process supports neuronal regeneration and enhances synaptic activity, ultimately helping to improve the pathology of AD and aid in the recovery of cognitive function [100]. PPARγ plays a central role in the regulation of glucose and lipid metabolism, particularly in adipose tissue. It is also involved in inflammatory response regulation [101]. In their research, Denner et al. found that the PPARγ agonist rosiglitazone improved the cognitive performance in AD model mice. This effect may be due to the way in which rosiglitazone overcomes the cognitive deficits and restores the neural network damage caused by AD through the extracellular signal-regulated kinase/mitogen activated protein kinase (ERK/MAPK) signaling pathway [102]. Recently, NRs have been shown to have direct neuroprotective effects in 5×FAD mice [103,104]. Mariani et al. reported that the RXR agonist bexarotene reduced neuronal loss, improved neuronal viability, increased the levels of presynaptic and postsynaptic markers, and enhanced cognitive performance in 5×FAD mice [103]. As shown in Figure 2, understanding the mechanisms of NRs in AD provides a helpful way to discover new targets for the treatment of AD.

## 4. Applications of NRs in AD

### 4.1. PPAR

PPAR is a transcription factor in the metabolic subfamily; it belongs to the phosphoproteins, whose transcriptional activity is affected by various kinases and phosphatases [105]. The main functions of PPAR are to regulate the uptake, transport, and metabolism of lipids and cholesterol [106]. PPAR is divided into three subtypes (PPARs): PPARα, PPARβ/δ, and PPARγ. The different subtypes of PPARs have different distribution in the body, and regulate different gene expression by binding to different reaction elements [106]. PPARs can regulate glycolipid metabolism, increase the expression of antioxidant enzymes, and provide neuroprotection by binding to the different ligands [107]. PPARs hold promise as a potential therapeutic target for treating AD, as PPAR agonists are commonly used in chronic neurodegenerative diseases and are widely expressed in the brain, especially in the hippocampus [108]. PPAR agonists have been shown to play an important role in the treatment of neurodegenerative diseases. In particular, PPARs can bind to the promoter region of the *β-site APP cleavase-1* gene, inhibit the expression of this gene, and thus inhibit the deposition of Aβ. Therefore, the activation of PPARs in neurons can also induce the clearance of Aβ [109].

It has reported that PPARα is widely expressed in the liver, kidney, and skeletal muscle and is involved in lipid metabolism to improve synaptic plasticity in AD mice [110]. Moreover, PPARα binds to cAMP response elements and directly regulates related hippocampal functions [111]. Roy A et al. applied immunohistochemical analysis and found that PPARα protein was located in the hippocampus; in particular, it was located in the CA1, CA2, CA3, and DG subregions of the hippocampus, which regulate the long-term learning and memory ability of mice [112]. PPARα plays an important role in fatty acid oxidation, lipid metabolism, and peroxisome proliferation [113]. Notably, Roy A et al. applied computer simulation, site-directed mutagenesis, and other technologies and found that statins were the ligands of PPARα [114]. After binding, statins can upregulate the neurotrophic factors in nerve cells and the brain and inhibit the regulatory enzyme HMG-CoA reductase in the cholesterol biosynthesis pathway [114]. Therefore, it reduces the cholesterol level, increases the hippocampus and cortical brain-derived neurotrophic factor, and improves the learning and memory ability of AD mice [114]. Furthermore, PPARα agonists also inhibit neuroinflammatory responses, amyloid deposition, and the activation of microglia and astrocytes in the hippocampus and cortex, and improve spatial learning and memory [115]. In another study, when Aβ was analyzed in hippocampal and cortical tissues, it was found that PPARα agonists, such as gemfibrozil and Wy14643, helped stabilize the expression of the human APP mutant. These agonists also reversed memory deficits in *APP/PSEN1ΔE9* mice by modulating the PPARα-mediated autophagy and lysosomal pathways [116]. In summary, PPARα can be explored as a new target for the development of AD diagnosis approaches and clinical therapy.

PPARβ/δ, a member of the steroid hormone ligand-inducible transcription factor superfamily, is widely expressed in vivo, especially in the dentate gyrus/CA1 region of the central nervous system [117]. It is a regulatory target of non-steroidal anti-inflammatory drugs, regulating brain glucose and cholesterol metabolism levels [118]. A study reported that the PPARδ agonists L-165041 and GW501516 exerted anti-apoptotic effects in vitro and emerged as drug candidates against neurodegenerative diseases [119]. A previous study reported that the PPARδ agonist GW0742 could significantly ameliorate the memory deficits, neuroinflammation, and apoptotic responses in mice [120]. In a clinical trial aimed at evaluating the safety, pharmacology, and metabolic effects of PPARδ agonists in patients with mild-to-moderate AD, the participants were treated with T3D-959 for 14 days. Systemic pharmacology was assessed through plasma metabolomics, while cerebral pharmacology was evaluated using FDG-PET scans to measure changes in the relative cerebral metabolic rate of glucose (RCMRgl) in AD-affected brain regions. Cognitive function was tested with ADAS-cog11 and the digit symbol substitution test (DSST) before, immediately after, and one week following treatment. The results showed that T3D-959 was generally safe and well tolerated, with dose-dependent pharmacokinetics. Plasma metabolomics revealed dose-dependent changes in lipid metabolism and insulin sensitization. The FDG-PET scans showed that the drug had a dose-dependent impact on RCMRgl in brain regions affected by AD. Cognitive assessments (ADAS-cog11 and DSST) indicated improvements that were potentially linked to the *ApoE* genotype and the drug’s pharmacodynamics, which were related to its mechanism of action. These findings suggest that the PPARδ agonist T3D-959 can modulate cerebral glucose metabolism and improve brain function in a dose-dependent manner [121]. Khorasani A et al. investigated the protective effect of abscisic acid on AD rat models induced by streptozotocin central injection and found that the effect of abscisic acid in combating learning and memory deficits in the AD rat model was significantly inhibited by PPARβ/δ antagonists [122]. The above studies suggest that PPARβ and PPARδ can be involved in AD prevention and treatment by multiple pathways and mechanisms.

PPARγ is widely expressed in adipose tissue and is the most studied isoform of PPARs; it is involved in the regulation of the inflammatory response, lipids, and carbohydrate metabolism in vivo [123,124]. Researchers have explored the relationship between PPARγ and AD at the genetic level [115,116,117]. For example, it was found that sortilin-related receptor 1 gene 1 (SORL1) gene variants were associated with increased AD risk [125]. Zhang et al. have investigated the impact of *SORL1* and *PPARγ* gene single-nucleotide polymorphisms (SNPs), gene–gene and gene–environment interactions, and haplotypes on late-onset Alzheimer’s disease (LOAD) risk [126]. The results indicate that the minor alleles of rs1784933 and rs1805192, as well as the gene–gene interaction between rs1784933 and rs1805192, are linked to an increased risk of LOAD. Additionally, a gene–environment interaction between rs1784933 and alcohol consumption, as well as a specific haplotype containing the rs1784933-A and rs689021-C alleles, is also associated with a higher risk of LOAD [126]. Moreover, researchers have explored the relationship between the *APOE ε4* allele and the risk of AD occurrence by examining 352 patients with genetically related late-onset AD versus 438 people labelled with the *APOE ε4* haplotype. The results revealed that *APOE ε4* allele non-carriers showed stronger anti-AD effects and that PPARγ genetic variation may alter the risk of AD occurrence in an *APOE ε4* allele-dependent manner [127,128].

Studies have shown that PPARγ plays a key role in regulating Aβ [101,129]. PPARγ influences the post-transcriptional regulation of APP, which is the precursor molecule that eventually produces Aβ. By affecting how APP is processed and how Aβ is cleared, PPARγ is thought to be involved in the development of AD. Additionally, PPARγ’s role in regulating inflammation and lipid metabolism may further contribute to its involvement in disease progression [75]. Medrano-Jiménez E et al. investigated the anti-inflammatory effects of mallow extract in familial AD mice by applying GW9662, a PPARγ inhibitor, as a control [101]. The results showed that the mallow extract was effective in reducing astrocyte proliferation, Aβ deposition, and learning memory deficits, and this effect was attenuated by the PPARγ inhibitor. Collectively, PPARγ has shown a positive effect in the treatment of AD.

### 4.2. Estrogen Receptor (ER)

Estrogen is produced by the enzyme aromatase in the ovaries, and its production is regulated by different life stages [130]. In the brain, estrogen signaling is essential for cognitive function and supports various processes, like neuron survival, growth, repair, and synaptic plasticity [131]. Estrogen receptors are found in key brain areas involved in memory, learning, and emotions, such as the hippocampus and prefrontal cortex [132]. According to the statistics, the risk of late-onset AD in women is twice that of men [132]. Notably, estrogen receptor α (ERα), estrogen receptor β (ERβ), and G-protein-coupled estrogen receptor 1 (GPER-1) together mediate estrogen action in vivo and play a key role in the hippocampus. It has been shown that loss of estrogen activity causes an increased incidence of neuronal death due to the production of Aβ plaques [133]. In detail, the loss of estrogen activity leads to increased neuronal death, primarily because estrogen helps protect neurons from damage, supports their survival, regulates the clearance of toxic proteins like Aβ, and maintains brain plasticity and repair mechanisms [134]. Without estrogen, these protective processes are compromised, which can lead to neuronal dysfunction, damage, and ultimately neuronal death [135]. This is a key factor in the increased risk of cognitive decline and neurodegenerative diseases such as AD in postmenopausal women. The ovaries affect brain health primarily through the production of estrogen and progesterone, which regulate key processes such as cognition, mood, synaptic plasticity, and neuronal repair [136]. These hormones help to maintain brain function, protect against neurodegenerative diseases, and promote emotional well-being. Hua et al. reported that a reduction in estrogen was associated with neurodegenerative pathologies in de-ovulated animal experiments. It was found that the de-ovulated AD rats had an accumulation of Aβ, an increase in the production of ROS, a decrease in ERα and ERβ, and neuronal death in their brains [133]. Tamoxifen is a medication commonly used in the treatment of breast cancer, particularly estrogen receptor-positive (ER-positive) breast cancer [137]. It is classified as a selective estrogen receptor modulator. Tamoxifen works by binding to estrogen receptors on cancer cells, blocking estrogen from binding and inhabiting cancer growth [138]. However, a study has explored the association between tamoxifen use and AD in aged women with breast cancer in Taiwan, finding that tamoxifen use was associated with increased odds of developing AD [139]. This effect may be due to the survival effect, not the toxic effect. Therefore, some ER agonists, such as tamoxifen and raloxifene, play a complex role in AD, and this requires further investigation.

### 4.3. Liver X Receptor (LXR)

Changes in cellular lipid metabolism that may affect APP processing have received widespread attention [140]. LXR, a ligand-activated transcription factor, is an endogenous signal that can affect AD [141]. A study has shown that the loss of natural LXR signaling in APP/PS1 mice leads to an increase in the accumulation of senile plaques [142]. Additionally, it was demonstrated that LXRs were powerful inhibitors of glial cell responses to inflammatory stimuli, such as Aβ. LXRs’ anti-inflammatory effects may enhance the ability of microglia to clear Aβ through phagocytosis in a chronic inflammatory environment. Therefore, LXRs regulate both cholesterol metabolism and inflammation, both in living organisms (in vivo) and in laboratory settings (in vitro), by reducing IL-1β levels and exerting anti-inflammatory activity [142]. Sun Y et al. [143] have verified that activation of LXR increases the lipid efflux, thereby decreasing Aβ production and exerting anti-AD effects. In an investigation of the effects of anti-inflammatory drugs on AD, Cui et al. [144] found that the activation of LXR significantly reduced validated markers in *APP/PS1* mouse brains by inhibiting the expression of NF-κB and decreasing the activation of microglia and astrocytes. Moreover, Koldamova et al. [145] explored the effects of the synthetic LXR ligand, T0901317, on AD in vitro and in vivo. The results showed that T0901317 increased the expression of ABCA1, a protein involved in cholesterol transport and reduced the ratio of soluble APP to its cleavage products. This reduction in the cleavage products led to decreased Aβ secretion, suggesting that T0901317 plays a role in modulating AD by influencing these processes. Therefore, LXR can be used as a potential target for the treatment of AD by regulating the lipid metabolism, reducing Aβ deposition in the brain, and improving the learning and memory ability of AD patients.

### 4.4. Farnesoid X Receptor (FXR)

FXR is widely expressed in the liver and intestine and plays an important role in the synthesis, secretion, and transport of bile acids [146]. Chen et al. [147] investigated the effect of FXR agonist 6ECDCA on AD, and found that the FXR mRNA and protein expression levels were enhanced in apoptotic neurons with excessive Aβ_1–42_ deposition in human neuroblastoma differentiated cells and mouse hippocampal neurons, respectively. Of note, FXR agonist 6ECDCA further enhances this effect, suggesting that FXR plays a potential role in the development of AD by regulating lipid levels.

### 4.5. Retinoid X Receptor (RXR)

There are three different isoforms of RXR: RXRα, RXRβ, and RXRγ; they are specifically expressed in different tissues and are involved in cell proliferation and differentiation as well as cholesterol metabolism [148]. Researchers have conducted a systematic screening of RXR gene variants in AD patients and individuals with non-demented Alzheimer’s neuropathology (NDAN) and examined the effects of gene polymorphisms with the development of AD and cholesterol levels. The results showed that the higher levels of 24s-hydroxycholesterol in the cerebrospinal fluid of NDAN patients may induce an increase in hepatic stanol synthesis, whereas cholesterol synthesis in the brain of AD patients may be reduced in the process of neurodegeneration. These results indicate that RXR gene variation affects the brain cholesterol metabolism and contributes to AD [149]. Furthermore, Dheer et al. [150] explored the effect of bexarotene, an RXR receptor agonist, on the level of RXR expression. It was found that RXR expression was reduced in AD mice and Aβ-treated cells, while bexarotene reversed the Aβ-induced deletion of RXR expression. Another study found that bexarotene induced the APOE-HDL particle production, reduced the microglia reactivity and plaque burden, and improved the cognitive function [151]. Surprisingly, the mice continued to show cognitive improvement and Aβ elimination within 2 weeks of discontinuation. These findings indicate that the RXR agonist can alter the brain Aβ pathology and cognitive changes [151].

RXR can act as a homodimer or heterodimer in vivo, and it has now been shown that RXR can form a heterodimer with VDR. It has been suggested that VDR deficiency is one of the potential factors in the development of late-onset AD [152]. In addition, mice with heterodimer and retinoic acid receptors (RAR)/RXR receptor mutations have been reported to have spatial learning and memory deficits [153,154]. These results all suggest that RXR is a potential target for the treatment of AD.

### 4.6. Aryl Hydrocarbon Receptor (AhR)

AhR is a ligand-activated cytoplasmic receptor and transcription factor that is widely expressed and functions in the central nervous system [155]. AhR is a multifaceted regulator with diverse functions in detoxification, immune system modulation, tissue development, metabolism, and even cancer [156]. Through its ability to respond to both environmental toxins and endogenous signals, AhR is involved in many critical physiological processes, and its dysregulation can contribute to various diseases, including cancer, autoimmune disorders, metabolic diseases, and neurodegenerative conditions [157]. Studies have shown that the AhR signaling pathway is associated with the antagonistic pleiotropy of brain aging as well as age-associated brain diseases [158]. Enkephalins is an endogenous catabolic enzyme of Aβ. Qian et al. [159] explored the role of AhR in Aβ metabolism, and found that the activation of AhR by endogenous or exogenous ligands significantly increased the levels of enkephalins in *APP/PS1* mice, which resulted in an improvement in the cognitive impairment and memory deficits of mice. Therefore, these findings provide a basis for the application of AhR agonist for the treatment of AD.

### 4.7. Other

In addition, there are many NRs involved in the occurrence and development of AD. For example, peroxisome proliferator-activated receptor- Gamma coactivator-1α (PGC-1α) is PPAR-γ coactivator-1α, a transcriptional coactivator that mediates many biological processes related to energy metabolism [160]. It controls energy metabolism mainly by acting on mitochondrial biogenesis and oxidative phosphorylation processes. Notably, it has been demonstrated that overexpression of PGC-1α results in an increase in mitochondrial DNA content [161]. A previous study compared PGC-1α overexpression mice with Eno2-cre-induced chronic cerebral under-perfused mice and found that PGC-1α expression was downregulated in chronic cerebral under-perfused mice. Furthermore, overexpression of PGC-1α significantly altered the cognitive function of mice, enhancing neuronal metabolic capacity and promoting neuronal activity under hypoxic conditions. The results suggested that PGC-1α could inhibit the overexpression of ROS levels and neuroinflammatory responses in the hippocampus of mice, positioning it as a potential target for the prevention and treatment of AD [162].

Additionally, it has been reported that the orphan NR Nurr1 is widely expressed in the central nervous system and plays an important role in the development of AD [163]. Qu et al. [77] used permanent bilateral common carotid artery occlusion (2-vo)2 to establish a rat model of chronic cerebral insufficiency of cerebral perfusion and found that after the injection of (2-vo)2, the rats showed cognitive deficits, and the activity of the hippocampal orphan NR LTX was reduced, which suggested that TLX had a protective role in the cognitive deficits caused by chronic cerebral perfusion. Additionally, PXR is a member of the orphan nuclear receptor subfamily, and the modulation of multiple transport proteins, such as P-glycoprotein, has been reported [164]. It has been reported that PXR is an effective way to improve central nervous system medication and eliminate Aβ in AD patients [165].

In addition to the above NRs, hormone-like receptors have also been reported to be involved in the progression of AD. For example, thyroid hormones are involved in central nervous system functioning and play an important role in brain plasticity and neurotransmission, and the hormone secretion is consistent with neuroblast proliferation [166]. Therefore, thyroid hormone receptor activity has been associated with the development of neurodegenerative diseases and psychiatric disorders [167]. Mineralocorticoid receptors are widely present in the brain, and the application of mineralocorticoid receptor antagonists in improving hippocampal learning memory has been demonstrated in the previous study [168]. Chen et al. [169] observed the therapeutic effects of mineralocorticoid receptor antagonists on Aβ-induced cognitive deficits in mice and found that mineralocorticoid receptor antagonism attenuated cognitive deficits and Aβ deposition and activated the nuclear factor erythroid 2-related factor 2 (Nrf2)-dependent antioxidant system. To sum up, NRs play a key role in the onset and progression of AD, and the development of new drugs targeting NRs may provide a new choice for the treatment of AD (Figure 3).

## 5. Conclusions and Perspective

In summary, NRs, including PPARs, ER, LXR, FXR, RXR, and AhR, are ligand-activated specific transcription factors that bind to DNA and activate or repress the transcription of the corresponding nuclear target genes and participate in the regulation of various biological processes. Of note, NRs are involved in the process of AD by improving Aβ deposition, regulating inflammatory response and body metabolism, and improving neuronal function. Therefore, NRs hold promise as a potential therapeutic target for treating AD.

NRs possess a wide range of functions and are involved in key processes such as inflammation, lipid metabolism, cell proliferation, and apoptosis. Consequently, numerous studies have been conducted, and several NR agonists, including PPARs, have been clinically applied. However, in the context of AD, the role of NRs is dual, as some NRs may exacerbate the progression of the disease. Therefore, the following issues also need to be resolved in the future studies: (1) It is important to identify the specific NRs that can help treat AD, while also avoiding the activation of certain NRs that could have harmful effects. (2) The structure of NRs can be studied in more detail using techniques like conformational searching, area calculations, and simulations based on molecular dynamics. These methods help in identifying potential drugs that target NRs and provide a deeper understanding of how these drugs interact with and affect NRs. (3) The mechanism of NRs participating in the AD process has not been thoroughly studied, and more attention should be paid to more biological processes, such as the tau protein formation and autophagy. It is believed that with the continuous understanding of AD, the role of NRs in AD will be clearer, and NRs will play a key role in the diagnosis, prevention, and treatment of AD in the future.

## Figures and Tables

**Figure 1 ijms-26-01207-f001:**
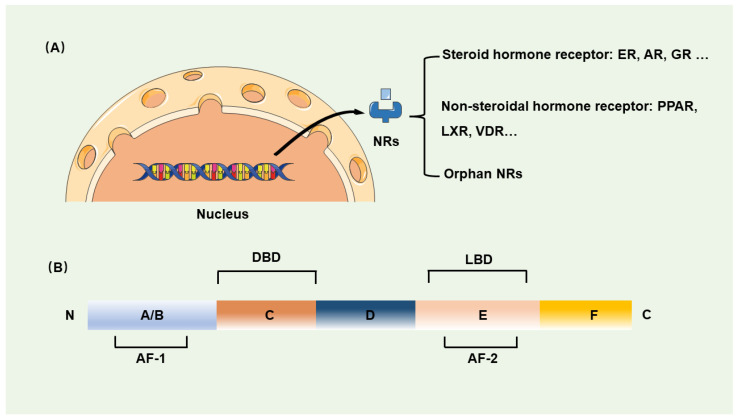
The classification of NRs (**A**) and their structures (**B**).

**Figure 2 ijms-26-01207-f002:**
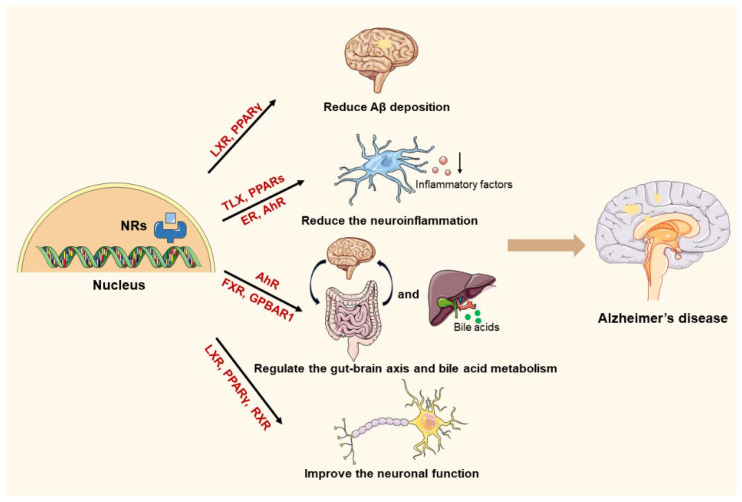
The mechanisms of NRs involved in AD.

**Figure 3 ijms-26-01207-f003:**
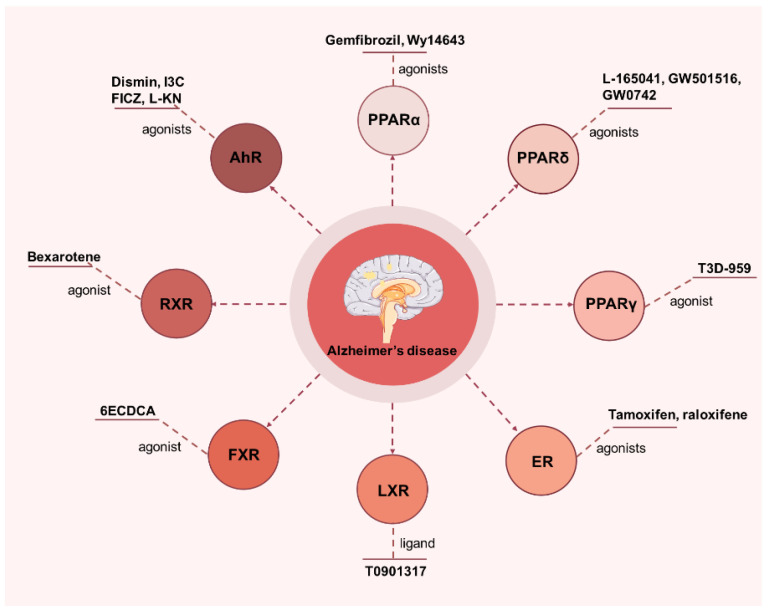
The applications of NRs in AD.

## Data Availability

No data were used for the research described in the article.

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
