# Peer review of "Recent Insights on the Role of Nuclear Receptors in Alzheimer’s Disease: Mechanisms and Therapeutic Application"

_ijms, 2025, doi:10.3390/ijms26031207_

Round 1
Reviewer 1 Report
Comments and Suggestions for Authors
The manuscript by Shan and Colleagues is potentially interesting as it addresses a relevant issue in the field of Alzheimer's disease research, i.e., the role of nuclear receptor in the onset and progression of disease. Any valuable information on this topic, possibly identifying potential pharmaceutical targets may indeed open new therapeutic strategies towards the cure of AD neurodegeneration. Despite the urgent need for a review on such topic, the work presented by the Authors is overall shallow, not well-organized, and poorly written. Therefore in my opinion the article is not suitable for publication in the present form.
General comments
This review article fails to provide proper description of experimental evidence concerning the topic. Manuscript structure is confused and headings/subheadings often misleading, not corresponding to the text contained therein. Throughout the manuscript, the Authors displayed mostly positive aspects of targeting NRs without addressing critical issues of these procedures. Moreover, they referred to either NRs’ agonists or antagonists, sometimes in a confusing manner. The manuscript lacks proper citations, thus failing to fulfill the purpose of a review article, which should instead provide a detailed and soundly based overview on the topic of interest.
Major concerns
The Abstract section focuses on the organization of the review, rather than providing a brief description of the topic. I would recommend a major remodeling of this section emphasizing general info on AD, NR and the common ground between the two.
The Introduction section should be revised in order to expand the considerations concerning AD and NR and why the latter is of interest in AD research. Specifically, the first part of the introduction should be rewritten to properly highlight AD pathogenesis (i.e. molecular hallmarks, sporadic forms and genetic mutations), rather than just mentioning some of the molecular features, such as ApoE mutations.
In "NRs as the therapeutic targets for AD" section, the main issue is that even though this section briefly talks about AD, most of the text has nothing to do with it; rather, it is a description of structural and functional activity of the NR taken into account. This section should be re-named.
In all the following sections ("The mechanisms of NRs in AD" etc.), what negatively impresses the reader is the lack of in-depth analysis of literature data. Considering that Aβ and neuroinflammation are immensely described in literature, the Authors should have described critical molecular events concerning such hallmarks rather than limiting to their relationship with NRs. "Regulating metabolism" section appears almost off-topic, at least in the present form.
"Improvement of neuronal function" section raises the question of when have the authors talked about neuronal dysfunction. Then how could readers understand functional improvement if the Authors have failed to provide them with information on AD features? This issue should be addressed.
In "PPAR" section, it is hard to understand why the Authors chose to underestimate the literature concerning the relationship between these receptors and AD. Some 592 results pop up when searching for PPAR & AD, impossible to summarize in less than 100 lines.
Minor concerns
The second part of the introduction concerns NRs and starts with this sentence:” In recent years, nuclear receptors (NRs) have attracted attention in the pathogenesis of AD [12, 13].” and literally has no general follow up: why were these of interest? The authors organized the writing so that the issue at hand could be understood only by researchers in the field.
On line 74, “2. Results” appears misplaced. Also, text format for this section is different from other sections: I would recommend text justification, as in other sections.
Line 212-213: “Currently, NRs have been shown to have direct neuroprotective effects in 5×FAD mice [58].”: one single citation and no explanation on neuroprotective NRs’ activity.
PPAR β/δ changes its name throughout the text (lines 267, 268) to PPAR δ/γ returning to its original name on line 273. Please correct or clarify the reason for this spelling.
More detailed comments are in the enclosed pdf copy of the article.

Author Response
|
Response to Reviewer 1 Comments
|
||
|
1. Summary |
|
|
|
Thank you very much for taking the time to review this manuscript. Please find the detailed responses below and the corresponding revisions/corrections highlighted/in track changes in the re-submitted files. |
||
|
|
|
|
|
2. Point-by-point response to Comments and Suggestions for Authors |
||
|
The manuscript by Shan and Colleagues is potentially interesting as it addresses a relevant issue in the field of Alzheimer's disease research, i.e., the role of nuclear receptor in the onset and progression of disease. Any valuable information on this topic, possibly identifying potential pharmaceutical targets may indeed open new therapeutic strategies towards the cure of AD neurodegeneration. Despite the urgent need for a review on such topic, the work presented by the Authors is overall shallow, not well-organized, and poorly written. Therefore, in my opinion the article is not suitable for publication in the present form. General comments This review article fails to provide proper description of experimental evidence concerning the topic. Manuscript structure is confused and headings/subheadings often misleading, not corresponding to the text contained therein. Throughout the manuscript, the Authors displayed mostly positive aspects of targeting NRs without addressing critical issues of these procedures. Moreover, they referred to either NRs’ agonists or antagonists, sometimes in a confusing manner. The manuscript lacks proper citations, thus failing to fulfill the purpose of a review article, which should instead provide a detailed and soundly based overview on the topic of interest. Major concerns Comments 1: The Abstract section focuses on the organization of the review, rather than providing a brief description of the topic. I would recommend a major remodeling of this section emphasizing general info on AD, NR and the common ground between the two. Response 1: We thank the reviewer for this comment. We have revised the abstract to emphasize the general information on AD, NR and the common ground between the two. Thank you again for your constructive comments. I believe these revisions will strengthen the abstract and make it more informative for the readers. Comments 2: The Introduction section should be revised in order to expand the considerations concerning AD and NR and why the latter is of interest in AD research. Specifically, the first part of the introduction should be rewritten to properly highlight AD pathogenesis (i.e. molecular hallmarks, sporadic forms and genetic mutations), rather than just mentioning some of the molecular features, such as ApoE mutations. Response 2: We thank the reviewer for this comment. We have rewritten the introduction section and marked red in the revised manuscript. Thank you again for your constructive feedback. I will ensure that the introduction clearly addresses both the molecular aspects of AD and the rationale for investigating NRs in the context of this neurodegenerative disease. Comments 3: In "NRs as the therapeutic targets for AD" section, the main issue is that even though this section briefly talks about AD, most of the text has nothing to do with it; rather, it is a description of structural and functional activity of the NR taken into account. This section should be re-named. Response 3: We thank the reviewer for this comment. We have changed this title to “Structure and function of NRs and their role in AD pathogenesis” in Line 119 and marked in red. Comments 4: In all the following sections ("The mechanisms of NRs in AD" etc.), what negatively impresses the reader is the lack of in-depth analysis of literature data. Considering that Aβ and neuroinflammation are immensely described in literature, the Authors should have described critical molecular events concerning such hallmarks rather than limiting to their relationship with NRs. "Regulating metabolism" section appears almost off-topic, at least in the present form. Response 4: We thank the reviewer for this comment. Firstly, we have added the in-depth analysis of literature data and marked in red. Secondly, we have supplied the description of critical molecular events concerning Aβ and neuroinflammation in the revised manuscript. Finally, metabolic dysregulation is increasingly recognized as a central feature in AD pathogenesis, contributing to the progressive neurodegeneration that underpins the disease. The brain is highly energy-demanding, and its proper function relies on efficient metabolism to support neuronal activity, synaptic plasticity, and cellular maintenance. In AD, however, several metabolic pathways are disrupted, leading to neuronal dysfunction, oxidative stress, and ultimately, cognitive decline. Understanding and regulating metabolism in the context of AD is critical for identifying potential therapeutic strategies to alleviate or slow disease progression. Therefore, the regulation of metabolism, including the regulation of the gut-brain axis and bile acid metabolism, is a key mechanism by which NRs are involved in AD. We have also emphasized the importance of metabolic regulation in AD in the revised manuscript. Comments 5: "Improvement of neuronal function" section raises the question of when have the authors talked about neuronal dysfunction. Then how could readers understand functional improvement if the Authors have failed to provide them with information on AD features? This issue should be addressed. Response 5: We thank the reviewer for this comment. We have added the information of neuronal dysfunction in AD in Line 343-355 and marked red in the revised manuscript. Comments 6: In "PPAR" section, it is hard to understand why the Authors chose to underestimate the literature concerning the relationship between these receptors and AD. Some 592 results pop up when searching for PPAR & AD, impossible to summarize in less than 100 lines. Response 6: We thank the reviewer for this comment. We agree that there are many study results about PPAR and AD. However, in this part, we focused on summarizing the application of PPAR in the treatment of AD. Therefore, we have screened the literature instead of citing all of them. Thank you again for your constructive feedback. Minor concerns Comments 7: The second part of the introduction concerns NRs and starts with this sentence:” In recent years, nuclear receptors (NRs) have attracted attention in the pathogenesis of AD [12, 13].” and literally has no general follow up: why were these of interest? The authors organized the writing so that the issue at hand could be understood only by researchers in the field. Response 7: We thank the reviewer for this comment. We have revised the description to “In recent years, nuclear receptors (NRs) have garnered significant attention in the study of AD pathogenesis due to their crucial role in regulating gene expression and cellular processes that influence neuronal health [12]. By modulating gene expression, metabolism, inflammation, and protein aggregation, NRs offer a promising therapeutic strategy for treating AD” in Line 82-85. Comments 8: On line 74, “2. Results” appears misplaced. Also, text format for this section is different from other sections: I would recommend text justification, as in other sections. Response 8: We thank the reviewer for this suggestion. We have deleted the wrong sections. Comments 9: Line 212-213: “Currently, NRs have been shown to have direct neuroprotective effects in 5×FAD mice [58].”: one single citation and no explanation on neuroprotective NRs’ activity. Response 9: We thank the reviewer for this comment. We have added the citation in Line 369. Additionally, the explanation of neuroprotective NR activity is as follows: “Mariani et al. reported that the RXR agonist bexarotene reduced neuronal loss, improved neuronal viability, increased the levels of presynaptic and postsynaptic markers, and enhanced cognitive performance in 5×FAD mice.” This revision is marked in red in Lines 369–372 of the revised manuscript. Comments 10: PPAR β/δ changes its name throughout the text (lines 267, 268) to PPAR δ/γ returning to its original name on line 273. Please correct or clarify the reason for this spelling. Response 10: We thank the reviewer for this comment. According to the literature [1], T3D-959 is a chemically unique, brain penetrant, dual PPAR delta/gamma agonist with 15-fold higher PPAR delta selectivity. Ubiquitous brain expression of PPAR delta, its critical role in regulating glucose and lipid metabolism, and the AD-like phenotype of PPAR delta null mice motivated this study. Therefore, we have changed “PPAR δ/γ” to “PPAR δ” in Line 435 in the revised manuscript. Comments 11: More detailed comments are in the enclosed pdf copy of the article. Response 11: We thank the reviewer for this comment. We have revised the article according to your suggestions one by one and marked the changes in red in the revised manuscript.
|
||
|
|
||
|
|
||
Reference
- Chamberlain, S.; Gabriel, H.; Strittmatter, W.; Didsbury, J., An Exploratory Phase IIa Study of the PPAR delta/gamma Agonist T3D-959 Assessing Metabolic and Cognitive Function in Subjects with Mild to Moderate Alzheimer's Disease. Journal of Alzheimer's disease : JAD 2020, 73, (3), 1085-1103.
Reviewer 2 Report
Comments and Suggestions for Authors
Review of “Recent insights on the role of nuclear receptors in Alzheimer’s disease: Mechanisms and therapeutic application” by Shan et al.
The exploration of nuclear receptors in Alzheimer’s disease (AD) is a timely and significant topic, especially since the last comprehensive review appeared in 2017. Many researchers are examining nuclear receptors as potential targets for AD therapy, and a clear overview of how these receptors might be involved in both pathogenesis and treatment would be highly valuable. Unfortunately, this review demonstrates multiple shortcomings and does not satisfactorily fulfill its stated goal.
The title promises insight into the “mechanisms” of nuclear receptors in AD. However, the review rarely explains how nuclear receptors contribute to AD. For instance, Section 3.3.1 (“regulating the brain–gut axis”) states that “AD may start in the gut and then spread to the brain,” yet offers no further explanation of the mechanism behind this claim. If the authors want to discuss mechanisms, they must delve into how nuclear receptors operate in these processes, not just mention associations or correlations.
A repeated sentence—“NRs are involved in the occurrence and development of AD”—appears multiple times (up to eight times) throughout the text. This repetition is not only redundant but also misleading; it implies that the research community broadly agrees nuclear receptors initiate AD, which is not an established consensus. If the authors hold that view, they need to provide robust evidence and rationale, acknowledging current debates on this issue.
Section 2, titled “NRs as the therapeutic targets for AD,” focuses predominantly on the chemical structures of nuclear receptors, rather than explaining why or how these receptors could be used therapeutically. Subsections often lack detail; some are so sparse that their inclusion seems unnecessary. When discussing potential therapeutic benefits of nuclear receptors, the review should summarize relevant findings and highlight gaps in knowledge that warrant further study.
Furthermore, the authors should address the field’s current consensus. For example, is there a specific receptor (such as PPAR or estrogen receptor) that stands out as particularly critical or promising for AD therapy? If so, why?
The abstract and the final paragraphs of the introduction present more like a procedural outline (“First we will do X, then Y, then Z”) rather than providing an informative summary of the review’s content and conclusions. Likewise, the conclusion fails to offer a clear stance. It claims that “some NRs may aggravate the progress of AD,” yet does not identify which receptors might do so, or why. This information could be crucial in guiding research.
Table 1 suffers from unclear purpose and excessive acronyms with no definitions or summary. If the authors cannot make this table more informative and self-explanatory, it should be removed.
Section 4.2 on the “Estrogen receptor” lacks precision. It posits that a reduction in estrogen is associated with neurodegeneration, then asserts that tamoxifen (an ER agonist/antagonist for breast cancer) reduces the risk of AD. Tamoxifen’s role is complex; it acts as an estrogen antagonist in breast tissue but can act as an agonist in other tissues. The review does not clarify why tamoxifen is relevant or whether the authors are suggesting it as a potential AD treatment.
Notably, Section 4’s structure—introducing a nuclear receptor’s basic biology followed by its possible application in AD—is an effective approach. However, the content needs to be clearer, especially when discussing studies or clinical trials. For example, in Section 4.1, “PPAR,” the review cites a clinical trial but focuses only on the drug’s safety profile without elaborating on efficacy or key endpoints. It would be useful to mention the trial’s objectives, what results were measured, and whether the intervention showed promise for AD.
Phrases such as “As you all know” appear throughout the paper. These should be removed, as they are informal and assume readers have specific knowledge. The review also contains grammatical and spelling errors, and, alarmingly, a leftover sentence from a template in the introduction instructs the authors on how to write results. Every paragraph requires careful editing to achieve clarity, logical flow, and precision.
In sum, extensive revision is needed. The review would benefit from:
- Clear explanations of mechanisms: If the authors state that nuclear receptors are involved in AD, they should specify how, why, and to what extent.
- Evidence-based conclusions: Statements about the role of nuclear receptors in AD onset or progression must be supported by research findings.
- Better organization: Each section’s title should match its content, and the text must flow logically from one idea to the next.
- Removal of redundancies: Repetitive or irrelevant text should be eliminated.
- Improved clarity on therapeutic applications: If this is an important focus of the review, more detail on how nuclear receptors might be targeted for AD therapy is essential.
- Polished writing: Grammar, spelling, and style should all be consistent and professional.
With substantial revisions, this review could become a helpful and informative resource for anyone interested in understanding nuclear receptors in Alzheimer’s disease.
Comments on the Quality of English Language
English writing of this manuscript requires significant attention and improvement.
Author Response
|
Response to Reviewer 2 Comments
|
||
|
1. Summary |
|
|
|
Thank you very much for taking the time to review this manuscript. Please find the detailed responses below and the corresponding revisions/corrections highlighted/in track changes in the re-submitted files. |
||
|
|
|
|
|
2. Point-by-point response to Comments and Suggestions for Authors |
||
Review of “Recent insights on the role of nuclear receptors in Alzheimer’s disease: Mechanisms and therapeutic application” by Shan et al. The exploration of nuclear receptors in Alzheimer’s disease (AD) is a timely and significant topic, especially since the last comprehensive review appeared in 2017. Many researchers are examining nuclear receptors as potential targets for AD therapy, and a clear overview of how these receptors might be involved in both pathogenesis and treatment would be highly valuable. Unfortunately, this review demonstrates multiple shortcomings and does not satisfactorily fulfill its stated goal.
General comments
The title promises insight into the “mechanisms” of nuclear receptors in AD. However, the review rarely explains how nuclear receptors contribute to AD. For instance, Section 3.3.1 (“regulating the brain–gut axis”) states that “AD may start in the gut and then spread to the brain,” yet offers no further explanation of the mechanism behind this claim. If the authors want to discuss mechanisms, they must delve into how nuclear receptors operate in these processes, not just mention associations or correlations.
Response: We thank the reviewer for this comment. We have added the explanation of how nuclear receptors operate in these processes in Line 290-310.
A repeated sentence—“NRs are involved in the occurrence and development of AD”—appears multiple times (up to eight times) throughout the text. This repetition is not only redundant but also misleading; it implies that the research community broadly agrees nuclear receptors initiate AD, which is not an established consensus. If the authors hold that view, they need to provide robust evidence and rationale, acknowledging current debates on this issue.
Response: We thank the reviewer for this comment. We have deleted the redundant expression and explained the reason of NRs are involved in the occurrence and development of AD.
Section 2, titled “NRs as the therapeutic targets for AD,” focuses predominantly on the chemical structures of nuclear receptors, rather than explaining why or how these receptors could be used therapeutically. Subsections often lack detail; some are so sparse that their inclusion seems unnecessary. When discussing potential therapeutic benefits of nuclear receptors, the review should summarize relevant findings and highlight gaps in knowledge that warrant further study.
Response: We thank the reviewer for this comment. We have revised the title to “Structure and function of NRs and their role in AD pathogenesis” and added the detail of on the chemical structures of nuclear receptors. When discussing potential therapeutic benefits of NRs, we have summarized relevant findings in Line 137-181.
Furthermore, the authors should address the field’s current consensus. For example, is there a specific receptor (such as PPAR or estrogen receptor) that stands out as particularly critical or promising for AD therapy? If so, why?
Response: We thank the reviewer for this comment. PPARs hold promise as a potential therapeutic target for treating AD, as PPAR agonists are commonly used in chronic neurodegenerative diseases and are widely expressed in the brain, especially in the hippocampus. We have highlighted this description and marked red in Line 385-388 and Line 631.
The abstract and the final paragraphs of the introduction present more like a procedural outline (“First we will do X, then Y, then Z”) rather than providing an informative summary of the review’s content and conclusions. Likewise, the conclusion fails to offer a clear stance. It claims that “some NRs may aggravate the progress of AD,” yet does not identify which receptors might do so, or why. This information could be crucial in guiding research.
Response: We thank the reviewer for this comment. We have rewritten the abstract and the conclusions and perspective sections.
Table 1 suffers from unclear purpose and excessive acronyms with no definitions or summary. If the authors cannot make this table more informative and self-explanatory, it should be removed.
Response: We thank the reviewer for this comment. We have deleted the table 1 in the revised manuscript.
Section 4.2 on the “Estrogen receptor” lacks precision. It posits that a reduction in estrogen is associated with neurodegeneration, then asserts that tamoxifen (an ER agonist/antagonist for breast cancer) reduces the risk of AD. Tamoxifen’s role is complex; it acts as an estrogen antagonist in breast tissue but can act as an agonist in other tissues. The review does not clarify why tamoxifen is relevant or whether the authors are suggesting it as a potential AD treatment.
Notably, Section 4’s structure—introducing a nuclear receptor’s basic biology followed by its possible application in AD—is an effective approach. However, the content needs to be clearer, especially when discussing studies or clinical trials. For example, in Section 4.1, “PPAR,” the review cites a clinical trial but focuses only on the drug’s safety profile without elaborating on efficacy or key endpoints. It would be useful to mention the trial’s objectives, what results were measured, and whether the intervention showed promise for AD.
Response: We thank the reviewer for this comment. In response, we have revised the description of tamoxifen's role in AD, and the changes are marked in red in Line 496–500. Additionally, further details of the study have been incorporated in Line 500–504.
Phrases such as “As you all know” appear throughout the paper. These should be removed, as they are informal and assume readers have specific knowledge. The review also contains grammatical and spelling errors, and, alarmingly, a leftover sentence from a template in the introduction instructs the authors on how to write results. Every paragraph requires careful editing to achieve clarity, logical flow, and precision.
Response: We thank the reviewer for this correction. We have corrected the writing mistake in the article and have thoroughly reviewed the entire manuscript for any other writing errors. All corrections have been marked in red in the revised manuscript.
In sum, extensive revision is needed. The review would benefit from:
Comments 1: Clear explanations of mechanisms: If the authors state that nuclear receptors are involved in AD, they should specify how, why, and to what extent.
Response 1: We thank the reviewer for this comment. We have included detailed explanations of the mechanisms and have marked the changes in red in the revised manuscript.
Comments 2: Evidence-based conclusions: Statements about the role of nuclear receptors in AD onset or progression must be supported by research findings.
Response 2: We thank the reviewer for this comment. We have revised the article according to your suggestions one by one and marked the changes in red in the revised manuscript.
Comments 3: Better organization: Each section’s title should match its content, and the text must flow logically from one idea to the next.
Response 3: We thank the reviewer for this comment. We have revised the title and ensured that the text flows logically from one idea to the next.
Comments 4: Removal of redundancies: Repetitive or irrelevant text should be eliminated.
Response 4: We thank the reviewer for this comment. We have deleted the repetitive or irrelevant text.
Comments 5: Improved clarity on therapeutic applications: If this is an important focus of the review, more detail on how nuclear receptors might be targeted for AD therapy is essential.
Response 5: We thank the reviewer for this comment. We have we have added details of the relevant study on therapeutic applications in AD and marked red in the revised manuscript.
Comments 6: Polished writing: Grammar, spelling, and style should all be consistent and professional. With substantial revisions, this review could become a helpful and informative resource for anyone interested in understanding nuclear receptors in Alzheimer’s disease.
Response 6: We thank the reviewer for this correction. We acknowledge the importance of maintaining consistent and professional writing throughout the manuscript. We have carefully revised the entire document to address grammar, spelling, and style issues to ensure clarity and precision. Additionally, we have made substantial revisions to improve the overall readability and organization of the review. We hope that these changes enhance the quality of the manuscript and that it will serve as a helpful and informative resource for researchers interested in the role of NRs in AD.
Comments on the Quality of English Language
English writing of this manuscript requires significant attention and improvement.
Response: We thank the reviewer for this correction. The revised manuscript has been polished by a native language editor and marked in red.

Round 2
Reviewer 1 Report
Comments and Suggestions for Authors
The Authors have substantially revised the manuscript entitled "Recent insights on the role of nuclear receptors in Alzheimer’s disease: mechanisms and therapeutic application", which now appears improved in its content, organization and clarity. Even the writing quality is higher, compared to the previous version. In my opinion, the paper is now suitable for publication.